# The Cryopreserved Sperm Traits of Various Ram Breeds: Towards Biodiversity Conservation

**DOI:** 10.3390/ani12101311

**Published:** 2022-05-20

**Authors:** Jakub Vozaf, Andrea Svoradová, Andrej Baláži, Jaromír Vašíček, Lucia Olexiková, Linda Dujíčková, Alexander V. Makarevich, Rastislav Jurčík, Hana Ďúranová, Peter Chrenek

**Affiliations:** 1Institute of Biotechnology, Faculty of Biotechnology and Food Sciences, Slovak University of Agriculture in Nitra, Tr. A. Hlinku 2, 94901 Nitra, Slovakia; xvozaf@uniag.sk (J.V.); jaromir.vasicek@nppc.sk (J.V.); 2NPPC, Research Institute for Animal Production Nitra, Hlohovecka 2, 95141 Lužianky, Slovakia; andrea.svoradova@mendelu.cz (A.S.); andrej.balazi@nppc.sk (A.B.); lucia.olexikova@nppc.sk (L.O.); linda.dujickova@nppc.sk (L.D.); alexander.makarevic@nppc.sk (A.V.M.); rastislav.jurcik@nppc.sk (R.J.); 3Department of Animal Morphology, Physiology and Genetics, Faculty of AgriSciences, Mendel University in Brno, Zemedelska 1, 613 00 Brno, Czech Republic; 4Department of Botany and Genetics, Constantine the Philosopher University in Nitra, Trieda Andreja Hlinku 1, 94974 Nitra, Slovakia; 5AgroBioTech Research Centre, Slovak University of Agriculture in Nitra, Tr. A. Hlinku 2, 94901 Nitra, Slovakia; hana.duranova@uniag.sk

**Keywords:** ram, sperm, cryopreservation, gene bank, biodiversity

## Abstract

**Simple Summary:**

Biodiversity protection is one of our most important challenges today. Local animal breeds, as an essential piece of cultural heritage, form integral part of the biodiversity of individual countries and regions. Cryopreservation of sperm is one of the powerful tools for the conservation of animal genetic resources by creation of gene banks containing long-term stored genetic material. Cryopreservation of ram sperm faces several obstacles, such as the possible influence of individuality and breed characteristics on resistance to damage caused by low temperatures. In our research, we deal with the investigation of these differences between various breeds bred in Central Europe.

**Abstract:**

The aim of our research was to compare three Slovak sheep breeds in the quality parameters of cryopreserved sperm. The ejaculates of Slovak Dairy (SD), Native Wallachian (NW), and Improved Wallachian (IW) sheep rams (n = 12) were collected by electro-ejaculation. Heterospermic samples were created from suitable ejaculates, separately for each breed (at least 90% of total and 80% of progressive motility). Samples were equilibrated in a Triladyl^®^ diluent and frozen by automated freezing. Sperm samples were subjected to the motility, morphology, (CASA), viability and apoptosis (DRAQ7/Yo-Pro-1), fertilizing capability (penetration/fertilization test (P/F) in vitro) and acrosomal status (transmission electron microscopy) assays before freezing and after thawing. It was found that there were no significant differences (*p* < 0.05) between the evaluated breeds in motility, viability, apoptosis, morphological properties, and fertilizing ability of cryopreserved sperm. Significant differences occurred in acrosomal status. Our results demonstrate that the use of the selected cryopreservation protocol is suitable for at least three different sheep breeds, which can greatly benefit the biodiversity protection and simplifies the creation of an animal genetic resources gene bank.

## 1. Introduction

Sperm cryopreservation is an important biotechnology tool for breed improvement for several species, including small ruminants. Efforts to optimize cryopreservation of ram sperm have increased in recent years in order to improve reproductive performance [1]. Although frozen and thawed ram sperm is a valuable tool for the distribution of genetic material, low pregnancy rates after an artificial insemination limit this technique. In addition to its use in the reproduction of livestock, cryopreservation is very useful and an effective approach in the animal genetic resources and biodiversity conservation [2]. It is used in order to facilitate long-term storage of biological material for later use or research. Additionally, the transport of genetic resources in the frozen state offers ethical and logistical advantages over the transport of live animals [3]. To date, it has not been possible to standardize the process of freezing and thawing to avoid the biochemical and structural damage in sperm cells (acrosome, nucleus, mitochondria, axoneme, and plasmatic membrane) that is produced by temperature changes, osmotic stress, and the formation of ice crystals, which cause oxidative stress and decrease in viability, motility and DNA integrity [4]. In particular, spermatozoa of small ruminants are extremely sensitive to cryopreservation compared to those of other species [5]. In addition to cell death, the surviving sperm cells may have damaged sperm organelles and membranes and reduced biological efficiency (e.g., sperm capacitation and acrosome reaction) [6]. Cryoinjury during freeze–thawing can be caused by factors such as thermal shock, ice formation, dehydration, increased salt concentration, and osmotic shock. Due to the aforementioned reasons, different diluents with cryoprotective substances have been studied and used to increase the sperm quality, prevent the formation of intracellular ice crystals, and reduce damage to the membrane during and after cryopreservation [7,8,9]. Some of the commonly used diluents for cryopreservation of small ruminant sperm are commercially available preparations such as AndroMed, Biladyl, Bioxcell, Optidyl, OptiXcell, or Triladyl [10]. Moreover, several diluents have been developed based on either egg yolk (citrate egg yolk-based diluent [11]), skimmed milk [5], or soybean lecithin [12,13,14,15,16,17,18] as a suitable egg yolk replacement and due to its undesirable antigenic properties [19]. Cryopreservation of ram sperm can be influenced and improved by the length of the equilibration time in both manual and programmable methods [20].

Native Wallachian, Improved Wallachian, and Slovak Dairy sheep are three of the five Slovak national sheep breeds. Native Wallachian sheep were brought to the territory of Slovakia in 13th or 14th century, so it is a very old and adaptable breed. Improved Wallachian sheep was generated by intentional combined crossing of Native Wallachian with rams of varied imported breeds (Texel, Hampshire, Cheviot, Leicester, and Lincoln) which improved the quality of wool and milk production. Slovak Dairy sheep was created by the crossing of the native breeds (Tsigai and Improved Wallachian) with specialized dairy breeds (Lacaune and East Friesian sheep) [21].

Cryopreservation has a negative effect on sperm cells, and differences can be observed between species, breeds, and individuals. The aim of our study was to examine the differences between the three above-mentioned Slovak sheep breeds on sperm quality parameters (motility, viability, apoptosis, morphological abnormalities, fertilizing ability, plasma membrane integrity, and acrosomal status) of fresh and mainly frozen/thawed (F/T) sperm.

## 2. Materials and Methods

### 2.1. Study Site and Animal Management

The research took place during the natural breeding season (September–November, 2020). The rams are kept in fenced enclosures, separated from females, in a local farm of National Agricultural and Food Centre—Research Institute for Animal Production, Nitra, Slovakia. Clinically healthy and sexually mature rams of Slovak Dairy, Native, and Improved Wallachian sheep breed (4 from each breed) were used in this experiment. The age of the individuals ranged from 2.5 to 5 years. The ambient temperature and humidity varied due to the external environmental conditions. The rams were fed with hay, mashed oats, commercially available BAK feeding mixture (0.75–1.2 kg daily per ram; BAK producer: PPD Prašice, Jacovce, Slovak Republic) consisted of organic wheat, organic barley, organic corn, and vitamin–mineral premix, and water was available ad libitum. The climate of the region is mild, with an average day temperature of about 15 °C and occasional rainfall at this time of the year.

### 2.2. Semen Collection and Processing

Semen was collected two times per week by electro-ejaculation (Electro-ejaculator, Minitube, Tiefenbach, Germany). The initial voltage was 0.5 V, then increased in each series of pulses (separated by 2 s breaks) up to a maximum value of 7 V. After reaching a voltage of 7 V, the pulses remained at this level until the end of ejaculation. For stress elimination, rams were treated with xylazine (0.2 mg/kg, Xylariem 2% average unit volume (a.u.v.), Riemser Arzneimittel GmbH, Greifswald, Germany) (if necessary). After collection, the sperm samples were transported to the laboratory in a collection vessel placed in a water bath at 30 °C. In the laboratory, semen was immediately evaluated for volume, concentration, motility (initial checking), and heterospermic samples representing each breed were created by mixing of suitable ejaculates (at least 90% of total and 80% of progressive motility). A total of 48 ejaculates (4 males, 3 breeds, 4 replicates) were used in our experiments. Heterospermic samples were created to avoid the effects of individual differences among rams. Aliquots were taken for fresh sperm analysis and the rest was diluted with Triladyl^®^.

### 2.3. Diluent Preparation and Freezing Process

On the day of collection, the final diluent was prepared by adding raw Triladyl^®^ (Minitube) (containing glycerol, Tris, citric acid, fructose, tylosin, gentamicin, lincomycin, and spectinomycin) to deionized water in a ratio of 1:3, then egg yolk was added to the final diluent to provide 10% *v*/*v*. After mixing the egg yolk, the mixture was filtered through filter paper. Then, each heterospermic sample was transferred to 15 mL test tube with prepared extender for dilution at ratio of 1:10 (semen:extender). The semen was diluted slowly with two or three gentle, vertical rotations at room temperature (RT). The 250 μL straws were subsequently filled with the diluted semen, sealed, placed in racks, and equilibrated in a refrigerator at 4 °C for 6 h [20].

After equilibration, straws were transferred to the pre-cooled automated IceCube freezing box (Minitube). The freezing program started automatically when the lid was closed; the program released nitrogen vapor into the box to provide the following temperature profile (+4 °C, −10 °C (120 s), −80 °C (450 s), −120 °C (100 s), and −140 °C (180 s). Following the stipulated time, the semen straws were plunged into liquid nitrogen in cryogenic container (Liquid nitrogen container, KGW-Isotherm, Karlsruhe, Germany) (−196 °C). After one week, samples were thawed by immersing the straws in a water bath (42 °C, 15 s), transferred into test tubes, and subjected for the F/T sperm analyses.

### 2.4. Sperm Motility Evaluation

The motility and sperm movement were analyzed by CASA (SpermVision^TM^ software, Minitube) with light microscope (at the 200× magnification; AxioScope A1, Zeiss, Tiefenbach, Germany) and Makler counting chamber (Microptic, Barcelona, Spain). Samples were diluted by saline (0.9% NaCl; Braun, Nuaille, Germany) at ratio 1:40 (*v*/*v*). A drop of diluted semen (10 μL) was transferred to a counting chamber and analyzed with manufacturer’s preset parameters for rams. For each sample, six microscopic view fields were analyzed, and the analysis itself was performed automatically. We mainly focused on total (TM) and progressive motility (PM). 

### 2.5. Morphological Changes

After motility analysis, an aliquot of samples was placed in the refrigerator and stored until the next day (approx. 24 h) in order to immobilize sperm for morphological analysis. Morphological abnormalities were measured by the same microscope as CASA was performed. In this analysis morphological abnormalities of sperm cells, e.g., detached flagellum from head, twisted flagellum, shortened flagellum, broken flagellum, cytoplasmic droplet flagellum, reduced or enlarged sperm head, or other pathological sperm were evaluated. The number of morphologically altered sperm with the total number of sperm (400) was compared and subsequently the percentage of individual sperm abnormalities was evaluated.

### 2.6. Viability and Apoptosis Assay

Aliquots of semen (1 × 10^6^·mL^1^ cells) were subdivided into prepared tubes intended for flow cytometric assessment of sperm viability and apoptosis. Briefly, apoptotic cells were stained with nuclear fluorochrome Yo-Pro-1 (100 μM; Molecular Probes, Lucerne, Switzerland). Samples were incubated at RT for 15 min in the dark. After incubation, the samples were centrifuged for 5 min (300× *g*) at 20 °C, the supernatant was carefully removed, and cells were washed with phosphate-buffered saline without Ca & Mg (PBS; Biosera, Nuaille, France). Subsequently, for the detection of dead sperm, a DRAQ7 fluorescent dye (1 μM; BioStatus Limited, Shepshed, UK) was used according to the manufacturer’s protocol. Unstained cells were determined as viable. The sperm samples were analyzed by a FACS Calibur flow cytometer (BD Biosciences, San Jose, CA, USA) and obtained data were evaluated with Cell Quest Pro^TM^ software (BD Biosciences) (Figure 1). At least 10,000 events (sperm cells) were analyzed in each sample.

### 2.7. Test of Sperm Penetrating/Fertilizing Ability

To evaluate the fertilizing ability of ram sperm samples, a heterologous penetration/fertilization test using bovine oocytes (n = 360; 120 oocytes for each breed) was performed. A heterologous IVF was used due to better availability of bovine oocytes compared to ovine ones. Ovaries were obtained at a local abattoir and subsequently transferred to the laboratory within 3 h at 25 °C for oocytes isolation. Briefly, isolated oocytes were matured during 23 h incubation in maturation medium E199 with glutaMAX (Gibco Invitrogen, Auckland, New Zealand) supplemented with gonadotropins (FSH/LH 1/1 I.U., Pluset, Lab. Calier, Barcelona, Spain), 0.25 mmol·L^−1^ of sodium pyruvate, 0.05 mg·mL^−1^ of gentamycin, and 10% of fetal calf serum (FCS, Gibco) at 39 °C with 5% of CO_2_ in the air. Matured oocytes were partially denuded of cumulus cells by 30 s vortexing. The zona pellucida was not removed as it is not required for IVF of bovine oocytes with ram sperm [22]. F/T sperm samples were used for IVF of oocytes. Fresh spermatozoa were used as a control.

The sperm suspension was washed twice using 200× *g* centrifugation in Sperm-TALP solution. The presumptive matured oocytes were then incubated in 100 μL drops of ferti-lization medium (IVF-TALP) under sterile mineral oil together with 4 × 10^6^ mL^−1^ ram sperm with the addition of 10 μg·mL^−1^ heparin (Sigma-Aldrich, Steinheim, Germany) and PHE solution (20 μM penicillamine, 10 μM hypotaurine, 1 μM epinephrine; Sigma-Aldrich) at 39 °C with 5% CO_2_ in air for 20 h.

Afterwards, the supposed zygotes were deprived of the excessive spermatozoa and/or residual cumulus cells by pipetting, washed twice in the PBS with 0.6% polyvinylpyrrolidone, covered with a drop of Vectashield anti-fade medium containing DAPI stain (Vector Laboratories, Inc., Burlingame, CA, USA), and mounted between coverslip and microslide. These preparations were stored in a fridge at 4 °C to await fluorescence analysis.

Zygotes after staining were monitored under a fluorescence microscope (Leica Mi-crosystems, Wetzlar, Germany) using blue fluorescence filter and 200× magnification objective. Sperm penetration into the oocyte ooplasm and creation of pronuclei were determined. The zygotes were considered penetrated when at least one sperm cell was observed un-der the zona pellucida. The zygotes were considered fertilized when two pronuclei, or one pronucleus and one sperm cell inside, were revealed. Non-penetrated oocytes showed no spermatozoa inside. Fertilizing capacity was determined as a ratio of penetrated + fertilized zygotes to total number of eggs.

### 2.8. Acrosomal Status

For assessment of changes caused by freezing, transmission electron microscopy was used as described previously by Olexikova et al. [23]. Briefly, fresh and F/T sperm samples were fixed in a fixative solution (2% paraformaldehyde and 2.5% glutaraldehyde in 0.15 mol·L^−1^ sodium cacodylate buffer, pH 7.1–7.3) during 1 h at 4 °C. Subsequently, the sperm was washed three times in cacodylate buffer for 15 min. Sperm pellets were post-fixed in 1% osmium tetroxide in cacodylate buffer during 1 h and embedded into 2% agar. Samples were then dehydrated by passing them through an acetone series and embedded into PolyBed resin (Polysciences Inc., Warrington, PA, USA). Ultrathin sections (70 nm) were placed on nickel grids, contrasted with uranyl acetate and lead citrate, and examined on a transmission electron microscope (JEM-2100, JEOL, Tokyo, Japan) operating at 80 kV. For each group, electronograms were made at the magnification of ×10,000. Randomly selected view fields were evaluated on multiple meshes with ultrathin section. From each group, at least 400 individual sperms were evaluated, cut through the front part of the head with visible acrosome. The sperms with a visible morphological abnormality were excluded from the evaluation. According to the state of the plasma membrane, the sperms were classified into three classes of quality. The first quality class included sperms with intact plasma membrane in the acrosome area and in the post-acrosomal region or with a slightly waved membranes present only in the post-acrosomal region. The acrosome mass was markedly dense (Figure 2a). Sperms with a slightly disturbed plasma membrane in the acrosome area as well as in the postacrosomal region were included in the second quality class (Figure 2b). The acrosome showed signs of enlargement and swelling. The third quality class included sperms with a significantly disturbed and damaged plasma membrane. Sperms exhibited loss of acrosomal content (Figure 2c).

### 2.9. Statistical Analysis

Experiments on cryopreservation of all three breeds were performed four times. For data normality distribution, the Shapiro–Wilk test was used. Subsequently, acquired data were evaluated by two-way ANOVA (Tukey method) using GraphPad Prism version 9.0.0 for Windows (GraphPad Software, San Diego, CA, USA). The data are represented as means ± SEM. Penetration/fertilization tests were performed in three replicates. Differences in the distribution of oocytes into categories of penetrated, fertilized or non-penetrated ones among breeds were analyzed by Pearson´s Chi-square test of independence (SigmaPlot 11.0, Systat Software Inc., San Jose, CA, UK). Differences at *p* < 0.05 were considered statistically significant.

## 3. Results

### 3.1. Sperm Motility Evaluation

No significant differences (*p* > 0.05) in the motility parameters were observed among the breeds, thus indicating that there is no effect of the breed properties on motility of cryopreserved sperm. A significant difference (*p* < 0.05) was observed only between the fresh and F/T groups in TM as well as PM in all breeds (Figure 3), most probably caused by the cryopreservation process.

### 3.2. Morphological Changes

Percentage of morphologically abnormal sperm after cryopreservation was below 15% in all the evaluated samples, which is in the range of commercial insemination dose standards (malformation rate ≤ 20%). No significant differences (*p* > 0.05) in the incidence of morphological abnormalities were observed between breeds (Table 1).

### 3.3. Viability and Apoptosis Assay

Based on the complexity of the sperm viability assessment, there was no significant difference (*p* > 0.05) in the proportion of viable, apoptotic, and dead spermatozoa among SD, NW, and IW. In the fresh and F/T groups, about 75% and 55% (respectively) of viable sperm were found in all breeds examined. In all three breeds, there was a significant difference (*p* < 0.05) between the fresh and F/T groups in the number of viable and dead sperm. (Figure 4).

### 3.4. Test of Sperm Penetrating/Fertilizing Ability

Penetrated oocytes were represented by MI and MII oocytes with at least one spermatozoon inside. Fertilized eggs were represented either by the egg with one pronucleus (PN) and at least one spermatozoon inside the zona pellucida, or by the zygotes with two or more PNs. Fertilizing ability did not differ among breeds, although the percentage of fertilized oocytes and the percentage of penetrated oocytes (64.52% and 3.22%) were numerically higher in the NW rams. (Figure 5).

### 3.5. Acrosomal Status

The effect of cryopreservation on acrosomal status of ram sperm from three different breeds is depicted in Figure 6. Freezing evidently resulted in sperm acrosome damage in all breeds. Among the three breeds, the acrosomal status of F/T sperm in the NW breed was significantly better compared to other breeds. There was a significant difference (*p* < 0.05) in the NW breed compared to SD and IW, where more intact acrosomes and less spermatozoa with released acrosome mass after freezing and thawing were found. In addition, this breed had the most sperm with intact acrosomes among the breeds, also in fresh samples.

## 4. Discussion

Conservation of farm animal genetic resources is an important challenge in order to maintain domestic biodiversity and adaptation of animal species to global changes, to breeding accidents or epidemics [24,25]. The genetic of many local or endangered breeds/species, with small population sizes, may be stored in the form of cryopreserved reproductive cells [26,27,28]. Ram sperm cryopreservation is less successful compared to other livestock such as bulls, rabbits, horses, etc. [29,30] due to higher sensitivity to freezing and thawing procedures and low cryotolerance of spermatozoa. In addition, individual variations in frozen sperm quality have been observed in small ruminants, suggesting specific differences in sperm susceptibility to freezing methods [31]. It is well known that native and purebred breeds are generally more vital and healthier than crossbreeds [32]. Therefore, it is important to investigate possible differences between the properties and quality of cryopreserved sperm of various breeds. Sperm assessment is important to determine the fertilization capacity of male livestock. Determination of motility and other microscopic parameters of sperm belongs to the main indicators of ejaculate quality. The CASA method provides a more detailed assessment of the basic parameters of sperm compared to visual and subjective assessments. By using devices belonging to the CASA system, it is possible to achieve objective results with sufficient repeatability [33].

In our research, we observed a relative decrease in total motility by 18.4% (SD), 15.1% (NW), and 15.2% (IW) and in the progressive motility by 27.5% (SD) and 25% (NW and IW) after freezing/thawing, compared to fresh sperm results. Galaraza et al. [34], using three different protocols for automated freezing of the Merino ram sperm, observed relative decrease in total motility by 30.1% (from 87.8% in fresh to 61.4% in thawed sperm) and in progressive motility—by 21.8% (from 34.8% in fresh to 27.2% in thawed sperm). Therefore, the decrease in total motility is thus significantly higher compared to our results. In progressive motility, the decrease was slightly lower, but this may be due to a lower value of progressive motility in fresh semen. In another study [35], Manchega ram semen was diluted with Biladyl^®^ extender (Minitube) and frozen by programmable biofreezer (Planer Kryo 10 Series III, Planer PLC, Sunbury-on-Thames, UK). There was a 49.1% relative decrease in total motility and 41.8% in progressive motility.

The DRAQ7/Yo-Pro-1 staining was applied for viability evaluation by flow cytometry. Following the cytometry procedure, sperm subpopulations were divided into three groups: (a) viable cells: negative for both DRAQ7 and Yo-Pro-1; (b) apoptotic cells: positive for Yo-Pro-1 but negative for DRAQ7; and c) dead cells: all cells positive for DRAQ7. A similar methodology was used by Masoudi et al. [36], with the difference that PI (Propidium iodide) instead of DRAQ7, and Annexin V instead of Yo-Pro-1 were used. In addition, the authors included early apoptotic sperm cells (Annexin V^+^, PI^−^) in the viable group. As there was no significant difference in the number of apoptotic sperm between the fresh and F/T groups, we can assume that the chosen cryopreservation method does not trigger apoptosis in the frozen sperm samples.

The state of the plasma membrane in front of the sperm head is closely related to the condition (protection) of the acrosome. The plasma membrane in the acrosome region has a different composition and properties (it is less stable) than in other parts of the sperm head to allow the acrosome reaction [37,38,39]. Fertilization is not possible without a functional acrosome and the course of the acrosome reaction. During freezing/thawing, the membrane is affected by several factors (disruption of the fluidity of membranes and structures during hypothermia, chilling injury, osmotic effects). When the plasma membrane is disrupted, the outer acrosome membrane is exposed, and the acrosome is compromised. Acrosomal integrity has been confirmed as one of the most important indicators of fertility in bulls [40]. Several authors have discussed the connection between the physical and chemical properties of the acrosome membranes on the sperm head and the functional properties of ram sperm. Parks and Lynch [41] pointed to an apparent relationship between sperm membranes glycolipid composition and cold shock. Additionally, recent work of Carro et al. [42] dealt with ram sperm cryosensitivity based on the lipid composition of their membranes and they conclude that the effective incorporation of cholesterol and desmosterol in ram sperm membranes has positive effect on biophysical behavior of membranes and improving functional parameters of sperm after temperature decrease and freezing. Glycolipids are localized in the sperm plasma membrane, which is the primary site of cold shock injury. Thus, different cold shock sensitivity and cryotolerance of different species or breeds may be related to individual lipid composition of sperm plasma membrane. This may be probably related to differences between breeds. In ultrastructural analysis in IW there were up to 14.4% of spermatozoa with membrane damage and reacted acrosomes (significantly more than in other breeds 6.4% SD; 5.7% NW). This could indicate individual or breed specific increased plasma membrane sensitivity. The results of the in vitro penetration test are also consistent with this hypothesis, indicating a trend (or tendency) of worst performance in IW (50%) versus 63.33% in SD and 67.74% in NW, although the results are not significant. On the contrary, the best result after IVF was obtained for the NW breed; the assessment of the ultrastructure of acrosomes in NW samples showed a higher proportion of intact acrosomes in both fresh and thawed samples and significantly less reacted acrosomes in thawed samples compared to other breeds.

## 5. Conclusions

Our results suggest that the cryopreservation regime, which was established in our recent research [20], is suitable for cryopreservation of three Slovak national sheep breeds, namely Slovak Dairy, Native Wallachian, and Improved Wallachian sheep. Parameters of motility, morphology, viability, and in vitro fertilization did not differ between individual breeds. However, significant differences among the breeds were observed in the acrosomal status revealed by transmission electron microscopy. Acrosome status is closely related to successful fertilization. The different effect of cryopreservation on the sperm acrosome of the examined breeds should be taken into account when establishing an animal gene bank. These findings could help to protect livestock biodiversity through possible reintroduction of native sheep breeds and long-term storage of male gametes as animal genetic resources.

## Figures and Tables

**Figure 1 animals-12-01311-f001:**
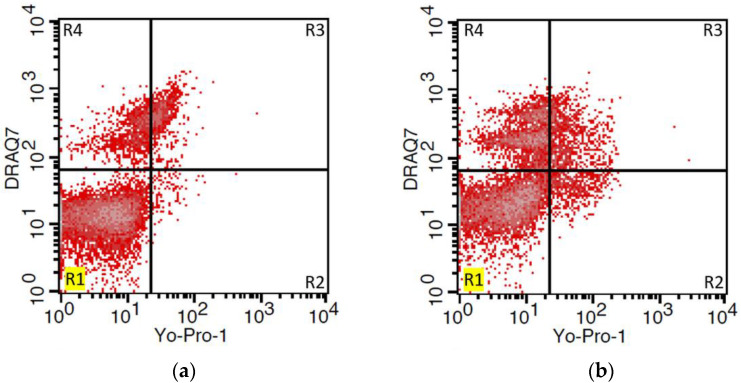
Expression of apoptotic and dead markers in fresh (**a**) and F/T (**b**) ram semen samples. Representative dot plots gated by regions showing the ratio of live sperm (unstained: Yo-Pro-1^−^/DRAQ7^−^) (R1), apoptotic (Yo-Pro-1^+^/DRAQ7^−^) (R2), and dead (Yo-Pro-1^+^/DRAQ7^+^ and Yo-Pro-1^−^/DRAQ7^+^) (R3, R4) sperm.

**Figure 2 animals-12-01311-f002:**
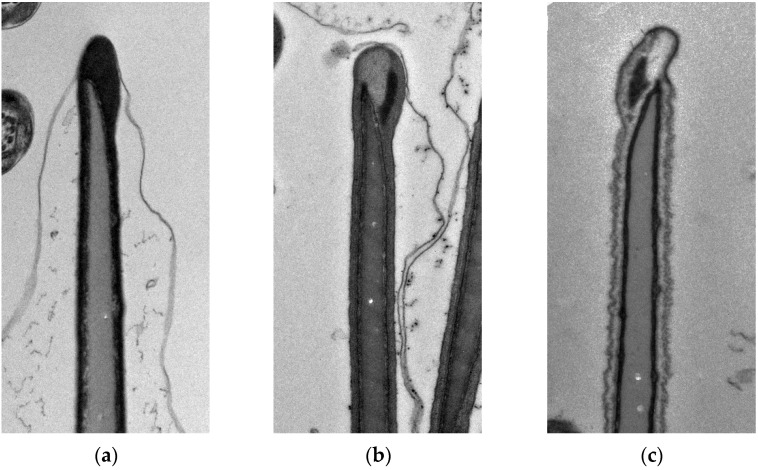
Sperm quality classes according to the state of plasma membrane. (**a**) intact acrosome, (**b**) swollen acrosome, and (**c**) damaged plasma membrane and released acrosome content.

**Figure 3 animals-12-01311-f003:**
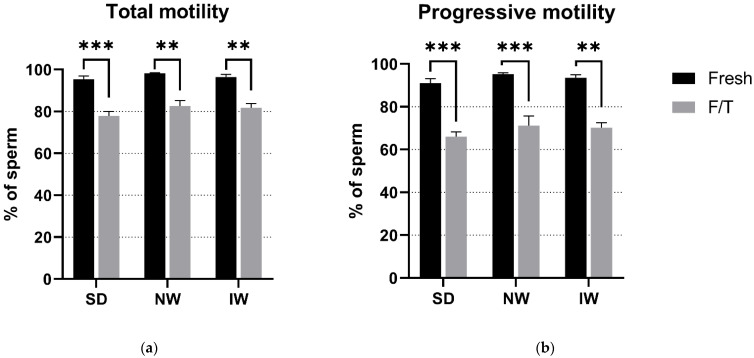
Comparative analysis of total (**a**) and progressive (**b**) motility among the breeds. SD—Slovak Dairy, NW—Native Wallachian, and IW—Improved Wallachian. No significant differences (*p* > 0.05) between breeds were found. ** (*p* < 0.01); *** (*p* < 0.001) Significant difference between the fresh and F/T groups. The data are represented as means ± SEM.

**Figure 4 animals-12-01311-f004:**
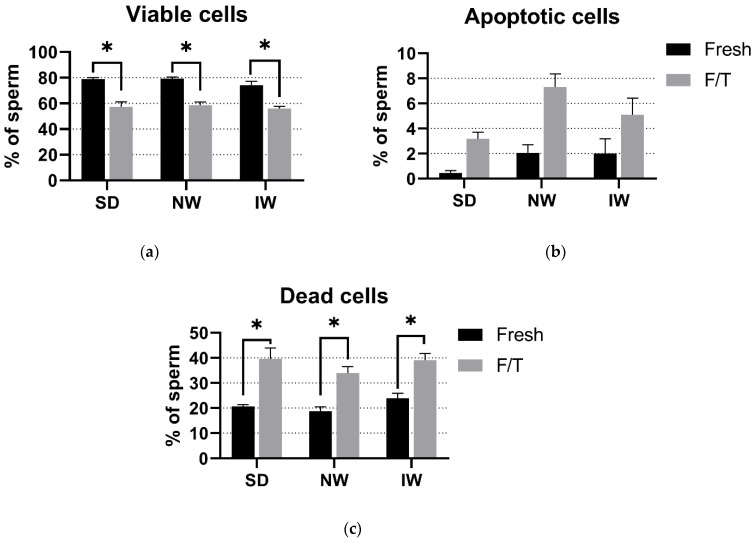
Comparative analysis of viable (**a**), apoptotic (**b**), and dead (**c**) spermatozoa among the breeds. SD—Slovak Dairy, NW—Native Wallachian, and IW—Improved Wallachian. No significant differences (*p* > 0.05) among breeds were found. * Significant difference (*p* < 0.05) between the fresh and F/T groups in the number of viable and dead sperm. The data are represented as means ± SEM.

**Figure 5 animals-12-01311-f005:**
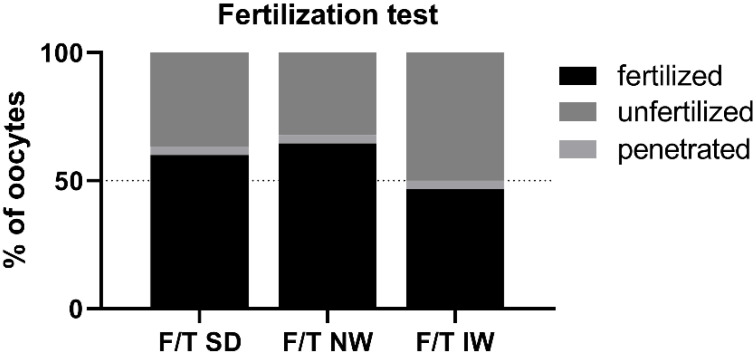
Penetration/fertilization test for each examined breed. F/T—frozen/thawed, SD—Slovak Dairy, NW—Native Wallachian, and IW—Improved Wallachian.

**Figure 6 animals-12-01311-f006:**
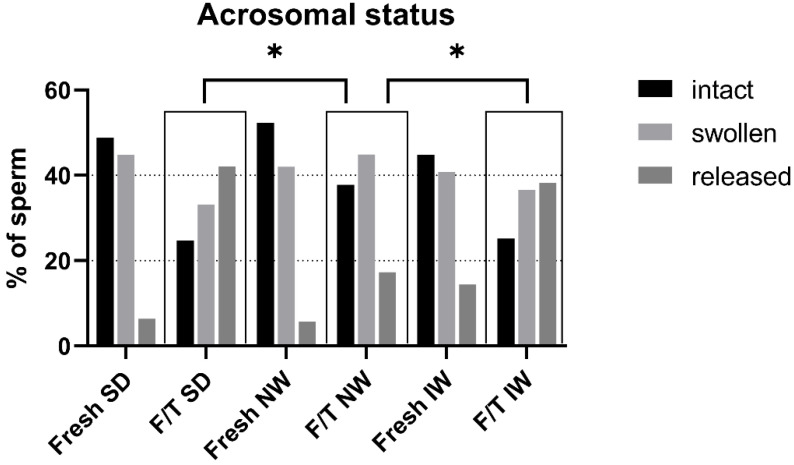
Acrosomal status of fresh and F/T sperm in each breed. F/T—frozen/thawed, SD—Slovak Dairy, NW—Native Wallachian, and IW—Improved Wallachian. * Distribution of sperm to the quality classes in F/T NW group differ significantly (*p* < 0.05) from distribution in F/T samples of other breeds.

**Table 1 animals-12-01311-t001:** Values of total and individual morphological abnormalities in fresh and F/T sperm for each examined breed (%).

Abnormality	SD	NW	IW
F-ST	2.0 ± 0.07	3.0 ± 0.2	2.0 ± 0.08
F-KT	3.8 ± 0.2	2.0 ± 0.16	3.0 ± 0.2
F-TT	1.0 ± 0.04	-	1.0 ± 0.04
F-RT	2.0 ± 0.06	3.0 ± 0.34	2.0 ± 0.1
F-BT	-	1.0 ± 0.1	-
F-CD	-	-	0.8 ± 0.04
F-Total	8.8 ± 0.4	9.0 ± 0.6	8.8 ± 0.5
F/T-ST	4.0 ± 0.17	5.0 ± 0.4	3.0 ± 0.07
F/T-KT	2.0 ± 0.08	2.0 ± 0.14	1.0 ± 0.05
F/T-TT	0.5 ± 0.05	1.0 ± 0.05	2.0 ± 0.1
F/T-RT	3.0 ± 0.2	3.3 ± 0.1	4.0 ± 0.2
F/T-BT	2.0 ± 0.15	2.0 ± 0.07	1.3 ± 0.1
F/T-CD	1.0 ± 0.03	1.0 ± 0.05	1.0 ± 0.05
F/T-Total	12.5 ± 1.0	14.3 ± 1.2	12.3 ± 0.8

F—fresh, F/T—frozen/thawed, ST—separated tail, KT—knob twisted, TT—torso tail, RT—rounded tail, BT—broken tail, CD—cytoplasmic drop, SD—Slovak Dairy, NW—Native Wallachian, and IW—Improved Wallachian. The data are represented as means ± SEM.

## Data Availability

The data presented in this study are available on request from the corresponding author.

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
