# Peer review of "The Cryopreserved Sperm Traits of Various Ram Breeds: Towards Biodiversity Conservation"

_animals, 2022, doi:10.3390/ani12101311_

Round 1
Reviewer 1 Report
My general remark is that the number of observations for each study is missing in the "Material and Method" and "Results" sections (on Figures and Tables). For example, how many zygotes have been tested by breeds? This should be included and the reader will be better convinced.
The processed literature seems relevant and sufficient. I suggest incorporating some recent literature on the efficacy of using fresh and F/T semen.
There is a high degree of identity in the genetic background of the three species. Do the authors think it is conceivable that the foreign breeds used in the crosses would be responsible for the significant difference (higher rate of plasma membrane disruption in IW, lower proportion of released F/T sperms in NW)? Please, write about it in the Discussion.
Detailed remarks:
Line 27 and 89: the number of rams was a total of 12 or 6 from each breed?
Line 28 and 106: how was the mixed sperm (heteroplasmic sample) made so that the individual effects were eliminated?
Line 28 and 37: “suitable protocol”. It is necessary to mention the extent to which the ejaculates taken did not meet the quality requirements (proportion of losses).
Line 30-31: the word "morphology" should be moved after "motility" because it was also evaluated by CASA, in concordance with 2.5. Morphological changes (Line 140).
Line 54: “oxidateve“ change for “oxidative”
Line 67: skim milk or skimmed milk (British English)?
Line 76-77: “Lacaune, East-Friesian Sheep” change for “Lacaune and East Friesian Sheep”
Line 85: “2.1. Study site and breeding condition”. It is not clear in the subchapter whether the rams were used in mating or lived separately from the females. If they had no contact with the females, the authors should explain the meaning of the word "breeding" in the title. If they had contact with females, was the effect taken into account when classifying sperm?
Line 87: of what abbreviation is “NPPC – RIAP”?
Line 157: if it is, I suggest emphasizing a heterologous in vitro fertilization (Ovis aries sperm x Bos taurus oocyte). A justification is needed as to why no homologous IVF was performed. Here, in this chapter, and not among the results (Line 281-282). Is (was) it necessary to remove the zone pellucida before heterologous fertilization?
Line 161: I suggest to revise “(0.25 mmol.l-1)
Line 162: I suggest to revise “(0.05 161 mg.ml-1)”
Line 169: I suggest to revise “106.ml-1”
Line 184-185: this sentence / calculation needs to be corrected because it is probably not the ovaries but the total number of oocytes that is the reference.
Line 217: “sperm” change for “sperms”
Line 218: “sperm” change for “sperms”
Line 220: “sperm” change for “sperms”
Line 221: “sperm” change for “sperms”
Line 226: “sperm” change for “sperms”
Line 248-249: “of three breeds” change for “of the three breeds”
Line 353: “sperm was diluted”. Sperm or semen was diluted?
Line 361-362: “In addition, he included early apoptotic sperm …” change for “In addition, they included early apoptotic sperms …”
Reviewer 2 Report
In this reviewer opinion, the manuscript is clear, generally well-written and organizated.
The statistical analysis should be critically revised, while some minor and major comments are proposed here below.
References are correct and well selected, but few. I would recommend to consider the works from Raspa et al (2017, 2018, 2021), which group has deeply analyzed sperm F and T (in mouse and after different freezing methods, but it could be interesting for the Authors).
Please, revise the whole manuscript for type errors and English grammar.
Major corrections:
The statistical analysis should be revised. For example, in Figure 2, why did you use a two-way ANOVA to compare the groups?? A two-way ANOVA is used to estimate how the mean of a quantitative variable changes according to the levels of two categorical variables. Here, you should compare just the control with the F/T group, every breed independently. By performing the statistical analysis in this way, the value of p is certainly very low, but not realistic, especially for the total motility. Please, re-analyse the data.
Same question for Figure 3. Please, re-analyse the experiments using the correct statistical test. The difference is very low.
Question: did you subject your data to normality test? the Authors did not specify that in the Statistical analysis section.
Table 2. Please, specify the significance
Regarding the acrosome status of the acrosome, and considering that spermatozoa after F/T could present some acrosome damage, this R would recomment to add some lines focused on the acrosome status, and in the fact that spermatozoa with a damaged acrosome could be suitable for IVF or artificial insemmination (two ways: apoptosis or fertilization). There are a lot of works regarding this aim.
Conclusions section is confusing and too long. Please, try to be more precise. This Reviewer could agree with the Authors in this conclusions if the statistical analysis are repeated and the significance is lower.
Minor corrections:
English spelling check is needed. Here some comments:
L48-49 "and an effective..."
L55 spermatozoa instead of spermatozoids
L59 " freeze-thawing, can be" remove the comma
L62 "sperm quality" instead of "spermatic quality"
L70 combination
L72 it is a very old
L95 this time of the year
L97 by an electroejaculation
L227 acrosomal instead of acrozomal
Figure 2: please add to the graph the column legends (black fresh, grey F/T).
L281: To evaluate.... instead of "for evaluation"
L361 "the Authors" instead of "he"
L390 please eliminate the first comma
Reviewer 3 Report
This manuscript gives a positive contribution to sperm quality evaluation (fresh vs frozen semen) in three Slovak breeds. However many faults have been detected namely ortographic and punctuation. Also methods are not detailed, so it is difficult to repeat and evaluate these experiments. In figures, i think that it is important to inserting significant differences (P<0.05)which will became easier , the understanding of this manuscript. Also the main fault is a defective description of results that have consequences in discussion and conclusions. I mean results are different only if P<0.05; otherwise they are not different. So it is essential to cortrect these errors, namely in results, discussion and conclusion. Also in tables we must insert different letters when P<0.05, and say in legend if it is in the same column or rows. Now i will touch some details.
In methods we need to indicate how much ejaculates were collected, number of sessions and the beginning and ending of this experiment. We . See lines 97-98. Also in the electroejaculation method it is essential to have intervals between electric pulses (see lines 99-100). In line 79, we need to include ....between species, breeds and individuals. In line 88, we see 6 rams from each breed and in line 27, we see 12. I am somewhat confuse. Many ortographic errors, , lines 54, 227 and 243. In method, we must detail in all techniques if we use neat or diluted semen, and semen was diluted with??; I am confuse with semen handling technique (section 2.2) and diluent preparation and freezing process ( section 2.3) , namely lines 108-112 vs 113-120. In section 2,2, . What is the sequence of semen evaluation?. Semen is extended with saline 0.09 NaCL (line 110) and extended with Triladyl (115-118). In all section we need to say if we use neat or diluted semen , and what is the extender, if it the case (see lines 133) and sections 2.5, and lines 147. Punctuation line, 164; Line 160, i think that it is a very high semen concentration for section 2.7; Fig 3, is incomplete, insert legend in Y axis. I think that result interpretation , lines 285-287, is not correct. . Say in table 2 if different letters are in rows or columns (P<0.05). . Complete fig 7, with insertion of significances, like in figure 2. In conclusion, line 388-389, is not correct (see P<0.05). So, due to my text, this manuscript, needs a major revision , before new submission.
Reviewer 4 Report
Here goes some points that I think may improve the comprehensibility of the paper.
Line 32: Plasma membrane integrity (PNA-FITC)
acrosomal integrity (PNA-FITC)
Lines 71-77: Before describing the origin of the Rams, It would be important to mention the reason why are you studying specifically with these three breeds.
Lines 80-81: In the quality parameters
On sperm quality parameters
Line 96: Collection and processing of semen
Semen Collection and Processing
Line 132: Makler counting chamber (Microptic, Spain)
(Sefi Medical Industries, Haifa, Israel)
Lines 138-139: stored until the next day.
¿How many Hours?
Line 150: RT
RT (Room Remperature)
Line 178: Leica fluorescence
¿Microscopic model?
Lines 186-204: Plasma membrane integrity
Spermatozoa are considered as viable when they have an intact plasma membrane. SYBR-14 is a fluorescent stain that can penetrate all cells, membrane damaged or intact, and bind to DNA. A dual stain of propidium iodide (PI) and SYBR-14 is commonly used to evaluate the viability of spermatozoa, while fluorescein isothiocyanate-conjugated peanut agglutinin (FITC-PNA), which specifically binds to the sugar Galactosyl β-1,3 N-acetylgalactosamine in acrosomal membranes, is used as a probe to visualize acrosomal integrity.
Line 199: ¿Why was the plasma membrana integrity (acrosomal integrity) not analised by flow cytometry or by fluorescence microscope?
Line 206: ¿Whats the meaning of TEM?
248: Figure 2. Percentage of motile (a) and progressively motile (b) sperm in fresh and F/T
Add a legend and color for each sample (fresh and F/T).
Line 376, 377: We found the most PNA-positive spermatozoa in IW (significantly more compared to other breeds) in the fresh.
This point need to be clarified
Round 2
Reviewer 4 Report
Line 201: For acrosome plasma membrane integrity the sperm samples were stained with fluorescently
Author Response
Line 201: For acrosome plasma membrane integrity the sperm samples were stained with fluorescently
- "For plasma membrane integrity" has been changed to "For plasma membrane integrity in acrosome area", as this was the purpose of this analysis.